# Impact of a Treatment Guide on Intravenous Fluids in Minimising the Risk of Hospital-Acquired Hyponatraemia in Denmark

**DOI:** 10.3390/jcm12155105

**Published:** 2023-08-03

**Authors:** Per Sindahl, Christian Overgaard-Steensen, Helle Wallach-Kildemoes, Marie Louise De Bruin, Kaare Kemp, Helga Gardarsdottir

**Affiliations:** 1Danish Medicines Agency, Division of Pharmacovigilance and Medical Devices, 2300 Copenhagen, Denmark; kkcmp@hotmail.com; 2Division of Pharmacoepidemiology and Clinical Pharmacology, Utrecht Institute for Pharmaceutical Sciences, Faculty of Science, Utrecht University, 3584 CG Utrecht, The Netherlands; m.l.debruin@uu.nl (M.L.D.B.); h.gardarsdottir@uu.nl (H.G.); 3Copenhagen Centre for Regulatory Science, Department of Pharmacy, Faculty of Health and Medical Sciences, University of Copenhagen, 2100 Copenhagen, Denmark; 4Department of Intensive Care 4131, Rigshospitalet, 2100 Copenhagen, Denmark; christian.overgaard.steensen@regionh.dk; 5Section for Social and Clinical Pharmacy, Department of Pharmacy, Faculty of Health and Medical Sciences, University of Copenhagen, 2100 Copenhagen, Denmark; helle.wallach.kildemoes.01@regionh.dk; 6Department of Clinical Pharmacy, Division Laboratories, Pharmacy and Biomedical Genetics, University Medical Center Utrecht, 3584 CX Utrecht, The Netherlands; 7Department of Pharmaceutical Sciences, School of Health Sciences, University of Iceland, 102 Reykjavik, Iceland

**Keywords:** risk minimisation, hyponatraemia, fluid therapy, intravenous fluids, drug regulation, patient safety, medication errors, prescribing practice, pharmacovigilance

## Abstract

Hypotonic intravenous (IV) fluids are associated with an increased risk of hospital-acquired hyponatraemia, eventually leading to brain injury and death. We evaluated the effectiveness of a treatment guide to improve prescribing practices of IV fluids. We conducted a before-and-after cross-sectional survey among physicians working at Danish emergency departments. The primary outcome was prescribing practices of IV fluids. Participants were asked which IV fluid they would select in four clinical scenarios. We applied multivariate logistic regression models to estimate the odds ratio of selecting hypotonic fluids. Secondary outcomes included knowledge about IV fluids and hyponatraemia, and the receipt, reading, and usefulness of the treatment guide. After the intervention, about a third (47/154) reported that they would use hypotonic fluids in patients with increased intracranial pressure, and a quarter (39/154) would use hypotonic maintenance fluids in children, both of which are against guideline recommendations. A total of 46% selected the correct fluid, a 3% hypertonic saline solution for a patient with hyponatraemia and severe neurological symptoms. None of the knowledge questions met the predefined criteria of success of 80% correct answers. Of the respondents, 22% had received the treatment guide. Since the implementation failed, we recommend improving distribution by applying methods from implementation science.

## 1. Introduction

Administering intravenous (IV) fluids containing water, electrolytes, and glucose is one of the most frequently used therapies provided in hospitals [1,2]. The primary goal of IV fluid therapy is to secure the circulating volume and to maintain the water and electrolyte balance [3]. Like other medicinal products, it carries a degree of risk. A risk that is often unrecognised because IV fluids is seen as routine, and is not regarded as a medicinal product with side effects [2,4,5].

In July 2017, the European Pharmacovigilance Risk Assessment Committee (PRAC) concluded that administration of hypotonic IV fluids is associated with an increased risk of hospital-acquired hyponatraemia, which in severe cases may lead to irreversible brain damage and even death [6]. Consequently, the PRAC updated the product information of hypotonic IV fluids containing electrolytes or carbohydrates (ATC code B05BA03, B05BB02 and B05BB01) to include warnings about hospital-acquired hyponatraemia [7]. It was furthermore agreed that additional risk minimisation measures such as educational material or direct healthcare professional communications should be decided and implemented nationally and be adapted to the national needs.

To investigate the national need for risk minimisation measures beyond the regular product information, the Danish Medicines Agency conducted a survey in 2019 [8]. About a quarter of the 201 responding health care professionals reported that they would use hypotonic fluids in patients with increased intracranial pressure, and 29.4% would use hypotonic maintenance fluids in children, both of which are against guideline recommendations [9,10,11,12,13]. Moreover, only 29.4% selected the correct fluid, a 3% hypertonic saline solution, for a patient with hyponatraemia and severe neurological symptoms, which is a medical emergency [10,14,15,16]. Hence, the Danish Medicines Agency concluded that further interventions to minimise the risk of hospital-acquired hyponatraemia were warranted in Denmark [8]. Subsequently, in June 2020, the Danish Medicines Agency decided together with the Danish Patient Safety Authority to distribute a treatment guide (pocket-folder) with basic information on IV fluids and hyponatraemia to all emergency departments in Denmark (see Appendix A).

The present study was conducted to evaluate the effectiveness of this treatment guide, implemented as part of risk minimisation measures to improve prescribing practice and increase knowledge about IV fluid treatment and hyponatraemia, thereby reducing the risk of hospital-acquired hyponatraemia.

## 2. Methods

The pre-defined study protocol is registered and accessible under the European Network of Centres for Pharmacoepidemiology and Pharmacovigilance (ENCePP), EU PASS Register No: 38732.

### 2.1. Study Design and Study Population

We conducted a survey to assess prescribing practices before and after distribution of a treatment guide aimed at minimising the risk of hospital-acquired hyponatraemia in Denmark. In this before-and-after cross-sectional study, the pre-measurement was performed in spring 2019 (from March through May) [8]. The treatment guide was distributed two times. First, in June 2020, treatment guides were sent to all hospitals with an emergency department, whereafter, in October 2020, treatment guides were sent to the same hospitals but addressed to the hospital management. The post-intervention survey started nearly one year after the first distribution of the treatment guide, from May through October 2021. The study population consisted of the target audience of the treatment guide, corresponding to physicians working in Danish emergency departments. We focused on emergency departments because IV fluid therapy is often initiated in the emergency department, and because inappropriate IV fluids prescribing has been reported as more likely to occur in emergency departments [17].

### 2.2. Recruitment

We invited physicians from all 38 emergency departments in Denmark, distributed across 21 hospitals [18], by mail to participate in the survey. In the case of no response to the first invitation, two more reminder attempts were made. To increase the response rate, we offered two options for filling out the questionnaire: it was either distributed via an online link by the head of the department or a paper version was distributed in person and completed during the daily meeting of the department. The online approach allowed respondents flexibility as to when they completed the questionnaire.

Participation was anonymous and voluntary. Consent for participation in this study was not needed according to the Danish Data Protection Authority.

### 2.3. Questionnaire

The questionnaire was developed by item generation through a review of the literature and the European Medicines Agency’s database of adverse drug reactions (EudraVigilance). To ensure the questions were representative of the concepts they were intended to reflect (hospital-acquired hyponatraemia), they were validated by a team of experts including an intensive care physician with extensive experience in fluid therapy and hyponatremia, pharmacovigilance officers and researchers in social pharmacy, pharmacoepidemiology and regulatory science.

Cognitive pretesting was performed on the questionnaire by four physicians to identify questions that were poorly understood, ambiguous, or which produced invalid responses. We used the ‘thinking aloud’ and ‘individual debriefing’ techniques. Questions for the ‘individual debriefing’ were prepared to search for potential problems not covered by the ‘thinking aloud’ technique [19,20,21]. Following the pretest and review, minor changes were made to reduce the number of questions and to increase the clarity and understanding of each question. Pretesting evidenced that it would take approximately 15 min to complete the questionnaire.

Both questionnaires (see Appendix A for an English version of the questionnaires), the pre-intervention and the post-intervention questionnaire, consisted of the following four parts:Background questions about hospital characteristics and physician characteristics.Clinical scenario questions (scenarios) based on real-life cases. Four scenarios describing different situations encountered in the emergency department in which IV fluids are administered were used to measure the prescribing practices of IV fluids.Eight knowledge questions. These were used to measure knowledge about IV fluids and hyponatraemia.Questions about the use of aids when prescribing IV fluids (e.g., treatment guidelines).

In addition to the above, the post-intervention questionnaire also included the following:5.Questions covering implementation measures (receipt and reading of the treatment guide) and usefulness of the treatment guide.

Due to concerns about dropout associated with the length of the questionnaire, five knowledge questions concerning prevention and treatment of over-correction that were part of the pre-intervention questionnaire were not included in the post-intervention questionnaire (see Appendix A). Only questions that were used in both surveys were used for the analysis of knowledge question.

All questions were closed ended (fixed alternative) with multiple-response choices, including an option of ‘do not know’.

The online questionnaire was designed such that it was not possible to navigate backwards and change previous answers. Furthermore, questions needed to be answered before the respondent could advance to the next question. We used LimeSurvey version 2.67.2 (Limesurvey GmbH, Hamburg, Germany, http://www.limesurvey.org, accessed on 20 January 2019, an open-source survey tool) for the electronic version of the questionnaire.

### 2.4. Outcome Measures

#### 2.4.1. Primary Outcome

The primary outcome measure was prescribing practice measured as incorrect selection of hypotonic fluids, as this is associated with hospital-acquired hyponatraemia [6]. To measure prescribing practices of IV fluids, we used the answers to the four scenarios in the questionnaire [22,23]. The scenarios were based on real-life cases in which IV fluids were administered, and describe conditions associated with increased antidiuretic hormone secretion where the risk of developing hyponatraemia has been well documented [6]. The scenarios covered:A high-risk (potentially increased intracranial pressure) patient with hypovolaemia.A child in need for maintenance intravenous fluids without hypovolaemia and hyponatraemia.A hypovolaemic and hyponatraemic (P-Na = 110 mmol/L) patient without severe symptoms of hyponatraemia.A hyponatraemic (P-Na = 118 mmol/L) patient with severe symptoms of hyponatraemia.

After each scenario, participants were asked to select the first-line treatment of choice from eight commonly used IV fluids in Denmark with different electrolytes and/or carbohydrates content and tonicity (see Table 1). In all scenarios, it was noted that the patient was not tolerating any oral intake (see correct answers to the questions in Appendix A).

The analysis of the primary outcome was based on respondents answering all scenario questions. For the scenario questions, the absolute risk of selecting hypotonic fluids was calculated. To decide whether further actions were warranted to protect patients from potential harms associated with the use of hypotonic IV fluids, we pre-defined an acceptable prescribing practice as a risk of being treated with hypotonic fluids after the intervention by no more than 10% of prescribers in each scenario.

#### 2.4.2. Secondary Outcomes

After being presented with the four scenarios, participants were asked to answer eight factual knowledge questions on the following topics:Renal water excretion in acutely ill patients.Intravenous fluids’ impact on P-Na in acutely ill patients.Hyperglycaemia and P-Na.Severe symptoms of hyponatraemia.Patients at high risk of severe symptoms.

In addition to the knowledge measures, secondary outcomes also included process measures (receipt and reading) and the usefulness of the treatment guide.

For the knowledge measures and implementation measures (receipt and reading), we used a pre-defined threshold of success of 80%, that is, at least 80% of respondents correctly answered each knowledge question and 80% of respondents received and read the treatment guide [24].

### 2.5. Data Analysis

We performed descriptive analyses on all primary and secondary outcomes, evaluating each question individually with respect to frequency and percentage.

We calculated the odds ratio (OR) of selecting hypotonic fluids before and after the intervention to estimate the impact of the intervention. We applied multivariate logistic regression models to estimate the OR with a 95% confidence interval (CI). For this analysis, we only included hospitals participating in both the pre-intervention and post-intervention survey. All background characteristics listed in Table 2 were individually assessed by means of univariate analysis and included in the multiple logistic regression when *p* < 0.10.

The Chi-squared statistic was used to test for differences in the pre- and post-intervention survey population.

All data handling and analysis was performed using SPSS (IBM Corp, released 2017, IBM SPSS Statistics for Windows, Version 25.0., Armonk, NY, USA).

Data were reported in aggregate format only, without any personal identifiers.

Missing data were not imputed.

## 3. Results

The primary analysis is based on respondents who responded to all scenarios (hereafter referred to as respondents). There were 201 respondents in the pre-intervention survey and 154 respondents in the post-intervention survey, corresponding to 55% (201/363) and 42% (154/363) of the total population of physicians working at emergency departments in Denmark, based on the estimated source population from 2014 [18].

### 3.1. Background Characteristics

Information about the physicians that participated in the pre-intervention and post-intervention survey can be found in Table 2.

Significant differences were found in characteristics for the sample that responded to the pre- and post-intervention surveys. Older physicians (>34 years), more senior (i.e., ended specialist training) physicians (44.8% vs. 55.8%), and those with more than five years of practice (60.2% vs. 75.3%) had a higher representation in the post-intervention survey sample of respondents compared to the pre-intervention survey. In addition, post-intervention survey respondents came from larger hospitals (39.3% vs. 62.3%), but fewer from hospitals with high complexity in terms of services they provide (30.8% vs. 20.1%) compared to pre-intervention survey respondents. Also worth noting is that fewer physicians from paediatric departments (50.2% vs. 35.7%) and more from trauma centres (1.0% vs. 13.0%) participated in the post-intervention survey compared to the pre-intervention survey.

### 3.2. Primary Outcome: Prescribing Practice

After the intervention, about a third (47/154) of respondents reported that they would use hypotonic fluids in patients with increased intracranial pressure (scenario 1), and a quarter (39/154) would use hypotonic maintenance fluids in children (scenario 2), both of which are against guideline recommendations (Table 3) [9,10,11,12,13]. Scenarios 3 and 4 met the pre-defined criteria of success of no more than 10% risk of being treated with hypotonic fluids after the intervention; 5.8% (9/154) reported using hypotonic fluids in patients with hypovolaemia without severe symptoms of hyponatraemia (scenario 3) and 7.1% (11/154) would use hypotonic fluids in patients with severe symptoms of hyponatraemia (scenario 4) [25]. In scenario 4, describing a patient with hyponatraemia and severe neurological symptoms, which is a medical emergency [10,14,15,16], 46% selected the correct fluid, a 3% hypertonic saline solution.

For all scenarios together, 20.1% and 17.0% incorrectly selected a hypotonic fluid pre and post intervention, respectively.

When comparing the results from the pre-intervention and post-intervention survey for hospitals that participated in both surveys, incorrect use of hypotonic fluids decreased (i.e., adjusted OR < 1) for all scenarios, but no results were statistically significant.

### 3.3. Secondary Outcomes

#### 3.3.1. Knowledge of IV Fluids and Hyponatraemia

None of the knowledge questions met the predefined criteria of success of 80% correct answers after the intervention (range 10.8–77.6%) (Table 4). There was little difference in correct responses before the intervention compared to after the intervention; the sum of correct answers to all knowledge questions was 38.3% and 43.1% before and after the intervention, respectively.

Improvement was observed for Q11c about Ringer’s lactate’s impact on P-Na in the acutely ill patient and Q12 about hyperglycaemia and P-Na [26].

#### 3.3.2. Implementation Measures and Usefulness

The implementation measures did not meet the pre-defined threshold of success of 80%, as only 28 respondents (21.5%) received the treatment guide and 22 respondents (16.9%) read it, out of the 130 participants who answered the question.

Nearly 80% (22/28) of those who received the treatment guide also read it.

Almost three-quarters (15/21) of those who read it found it useful, and a third (7/21) reported that they changed their prescribing behaviour patterns after reading the treatment guide.

## 4. Discussion

In 2021, risk minimisation measures to reduce the risk of hospital-acquired hyponatraemia were implemented in Denmark, in the form of a treatment guide on hyponatraemia and IV fluids. To evaluate the impact of this treatment guide, we measured prescribing practice and knowledge of hyponatraemia and IV fluids before and after implementation of the guide.

### 4.1. Main Results and Lessons Learned

The main question addressed in this study is whether the treatment guide was effective at improving prescribing practice of hypotonic fluids that are associated with the risk of hospital-acquired hyponatraemia, that is, whether the treatment guide reduced the incorrect selection of hypotonic fluids in the four scenarios. When participants were asked which IV fluid they would use in four clinical scenarios, a decrease in incorrect use of hypotonic fluids was observed in three out of four scenarios after the intervention. However, since the implementation failed, with less than a quarter of the participants receiving the treatment guide, a causal association between improvement in prescribing practice and the introduction of the treatment guide was considered implausible. Overall, there was consistency between knowledge and prescribing practice, as there was little difference in knowledge before and after the intervention.

Evaluating the effectiveness of risk minimisation measures should also provide evidence to regulators to determine whether further interventions are needed. A priori, we defined a success criterion for the present intervention that no more than 10% of patients should be at risk of being treated with hypotonic fluids after implementation of the treatment guide. The success criterion was not achieved. Children and patients with potentially increased intracranial pressure are patient populations at particular risk of hyponatraemic encephalopathy upon hypotonic IV fluid treatment. Our results show that more than 10% incorrectly selected hypotonic IV fluids in these patient populations, resulting in an increased risk of hospital-acquired hyponatraemia. In addition, the demonstrated lack of use of hypertonic saline solution (54%) in patients with hyponatraemia and severe neurological symptoms is of concern, as this is a medical emergency requiring prompt adequate therapy. Hence, it is concluded that the treatment guide did not sufficiently improve prescribing practice, and consequently further initiatives are warranted to reduce the risk of hospital-acquired hyponatraemia due to inappropriate treatment with hypotonic fluids.

The most surprising result was the low receipt of the treatment guide (21.5%). The implementation failure may have several explanations, e.g., although delivered, the guide was not forwarded to the clinicians due to administrative staff filtering, and/or the target population work at other departments than the department to which the guide was sent (e.g., some hospitals provide care to children and adults in separate locations within a facility) [27]. The experience with this study demonstrated that distributing educational material to emergency departments addressed to the department or the hospital management without further specification is inadequate for reaching the target population, namely the treating physicians. Thus, the question is: How do we reach the target population?

### 4.2. Reaching the Target Population

Healthcare professionals are exposed to a vast amount of information on medicines from many different sources (e.g., marketing material from pharmaceutical companies, publications in peer-reviewed journals, guidelines) [28]. This means that reaching clinicians with information on medicines is challenging. According to a Cochrane review [29], printed educational materials, when used alone and compared with no intervention, have a small beneficial effect on professional practice outcomes. However, from the data gathered, they could not comment on which characteristic of the educational materials influenced the effectiveness. Simon and Morrato advocate for advancing the practice of risk minimisation by applying methods from the field of implementation science, i.e., the study of strategies to adopt and integrate evidence-based health interventions and change healthcare practice patterns [30]. According to implementation science, active dissemination efforts involving peer-to-peer human interaction have been proven to be more effective than passive strategies, such as printed pamphlets alone [28]. It is also recommended to engage and identify opinion leaders/idea champions within the local organisation and/or target audience to facilitate implementation [28,30,31]. Moreover, a combination of interventions, such as information dissemination supported by opinion leaders, continuing education events and educational outreach visits, followed up with reminders and participation options with self-directed behaviour change, are more likely to be successful [32].

### 4.3. Implications for Regulators and Lessons for the Future

If a risk minimisation measure proves ineffective (i.e., did not meet the pre-defined criteria of success), it is imperative that corrective or alternative measures are identified and implemented [21,33].

We evaluated different levels of the intervention: theoretical knowledge, prescribing practice, implementation and usefulness [34]. This approach is suitable for ascertaining whether the lack of success was caused by a failure in the implementation and/or the content/format of the treatment guide [33].

Our results suggest that if the clinicians receive the treatment guide, they will read it (80%), that most of those who read it perceived the treatment guide as useful (71%), and last but not least, that one-third reported having changed their prescribing behaviour after reading the guide. Although caution must be applied in the interpretation of our findings due to the small sample size (n = 22), our study indicates that the content and format of the treatment guide is useful for clinicians administering IV fluids. In fact, the treatment guide had been through an extensive review process by key stakeholders, including three authorities (the Danish Patient Safety Authority, Danish Medicines Agency, and Danish Health Authority), emergency physicians and five Danish medical associations: the Danish Endocrine Society, Danish Society of Anaesthesiology and Intensive Care Medicine, Danish Paediatric Society, Danish Geriatric Society and Danish Society for Emergency Medicine.

According to the systematic review by Artime et al. [35], regulators required re-distribution or improved distribution in 31.8% of effectiveness studies using a survey design because of low receipt of materials. Acknowledging that the implementation had failed, we will have to both improve the distribution and redistribute the treatment guide, taking into consideration the recommendation from implementation science. Risk management is an iterative process of evaluation, correction and re-evaluation of risk minimisation measures [34].

### 4.4. Strengths and Limitations

Our study has several strengths. We approached the complete source population (i.e., all physicians working at an emergency department), used a before-and-after design and pre-defined measures of success. According to systematic reviews of studies assessing the effectiveness of risk minimisation measures in Europe, pre-defined thresholds of success are rarely used, and none had a before-and-after survey design [35,36].

An additional strength is the use of clinical scenarios based on real-life cases to measure behaviour, i.e., prescribing practice. We thereby avoided the limitations of self-reported data, which is commonly used in surveys that measure behaviour [21,35].

The results of this study must be considered with several limitations in mind:(1)Although the response rate was moderate in both the pre- and post-intervention surveys (55% and 42% of the total population of physicians working at emergency departments in Denmark [37], respectively), compared to physicians’ general low response rate in surveys [35,38,39], the non-probability sample design may limit the generalisability of our results. However, our previous study did not find any predicting factors of prescribing practice [8]. Additionally, our results did not change after adjustment for imbalances of background characteristics. It is, therefore, unknown how the sample design may have influenced the generalisability of the results.To improve the response rate, we offered teaching in IV fluid and hyponatraemia after the post-intervention survey, as we experienced a lack of engagement for participation due to the extensive workload of the healthcare system following the COVID-19 pandemic.(2)In line with many surveys, this study is limited by selection bias, as participation is more likely among engaged and/or educated physicians than among those who did not respond [21].(3)Another potential bias is the artificial setting of the survey which differs from the clinical setting in the emergency department where you may be busy and distracted under clinical pressure. Taken together, the inherent biases of this survey tend to overestimate the prescribing practice of the physicians.(4)Extrapolation of the results is limited by the fact that the surveys were only administered to physicians working in Danish emergency departments.(5)Finally, it is evident that the objective of a risk minimisation measure is to reduce the frequency and/or severity of an adverse reaction [21,34]. This study did not explore the key outcome measure, hyponatraemia, as this was considered unfeasible due to a lack of IV fluid administration recording and due to the challenge in linking prescribing of hypotonic IV fluid treatment and the occurrence of hyponatraemia.

## 5. Conclusions

Overall, our study showed a slight decrease in incorrect use of hypotonic IV fluids after introduction of a treatment guide in Denmark aimed at improving prescribing practice, and thereby reducing the risk of hospital-acquired hyponatraemia, which is frequently caused by inappropriate administration of hypotonic IV fluids.

However, the pre-defined threshold of success for prescribing practice was not achieved in children and in patients with potentially increased intracranial pressure. Additionally, less than half of the respondents would use hypertonic saline solution in patients with severe symptoms of hyponatraemia, which is a medical emergency requiring immediate adequate therapy to prevent severe neurological sequelae. These findings call for further intervention.

The failed implementation may explain the insufficient impact of the treatment guide. Hence, we recommend the redistribution of the treatment guide and improving the distribution by applying methods from implementation science.

## Figures and Tables

**Table 1 jcm-12-05105-t001:** IV fluid response choices indicating sodium concentration and tonicity after injection.

Fluid Response Choices of the Scenarios ^1^	Sodium Concentration	Tonicity After Injection
Glucose 5% isotonic	0 mmol/L	Strongly hypotonic
Darrow–glucose	31 mmol/L	Strongly hypotonic
Potassium–sodium–glucose	40 mmol/L	Strongly hypotonic
0.45% NaCl with 2.5% glucose isotonic	77 mmol/L	Strongly hypotonic
Ringer’s acetate	130 mmol/L	Moderately hypotonic
Isotonic saline solution	154 mmol/L	Isotonic
0.9% NaCl with 5% glucose	154 mmol/L	Isotonic
3% NaCl	513 mmol/L	Strongly hypertonic

^1^ See Appendix A for a full description of the composition of the fluids.

**Table 2 jcm-12-05105-t002:** Background characteristics of physicians who responded to all scenarios in the pre- and post-implementation survey^1^.

			Pre-Intervention (N = 201)		Post-Intervention (N = 154)		χ^2^ Test
			*n*	(%)	*n*	(%)	*p*-Value
Characteristics of physicians							
Gender							0.52
	Female		119	(59.2)	91	(59.1)	
	Male		82	(40.8)	62	(40.3)	
Age							0.08
	18–34 years		80	(39.8)	44	(28.6)	
	35–44 years		54	(26.9)	53	(34.4)	
	≥45 years		67	(33.3)	57	(37.0)	
Number of patients treated with intravenous fluids weekly							0.14
	0 patients		28	(13.9)	10	(6.5)	
	1–5 patients		68	(33.8)	51	(33.1)	
	>5 patients		104	(51.7)	92	(59.7)	
Years of practice							0.007
	≤5 years		78	(38.8)	38	(24.7)	
	>5 years		121	(60.2)	116	(75.3)	
Position ^2^							0.04
	Junior doctor		111	(55.2)	67	(43.5)	
		FY1 ^3^	25	(12.4)	7	(4.5)	
		FY2 ^4^	26	(12.9)	19	(12.3)	
		Specialty registrar	47	(23.4)	39	(25.3)	
		Other	13 ^5^	(6.5)	-	-	
	Senior doctor		90	(44.8)	86	(55.8)	
Characteristics of EDs ^6^							
Size							<0.001
	Large		79	(39.3)	96	(62.3)	
	Medium		104	(51.7)	42	(27.3)	
	Small		18	(9.0)	15	(9.7)	
Complexity ^7^							0.04
	High		62	(30.8)	31	(20.1)	
	Medium		139	(69.2)	122	(79.2)	
Type							<0.001
	Combined general population ED ^8^		29	(14.4)	20	(13.0)	
	Adult ED		69	(34.3)	58	(37.7)	
	Paediatric ED		101	(50.2)	55	(35.7)	
	Trauma centre		-	-	20	(13.0)	

^1^ Cells where n < 5 are omitted/blank. This also includes missing/unknown information. ^2^ The terms ‘junior’ and ‘senior’ in the medical profession indicate whether or not a doctor is still in training and whether they can practice independently without supervision. A senior doctor has completed specialist training. ^3^ Foundation doctor year 1. ^4^ Foundation doctor year 2. ^5^ ‘Other’ includes medical students, unspecified junior doctors, pre-FY1, and PhD students. ^6^ Characteristics of the emergency departments (EDs) were based on a report of EDs by the Danish Ministry of Health [18]. ^7^ Complexity of services they provide. ^8^ Combined general population EDs provide care for all patients in one area, while separate general population EDs provide care to children and adults in separate locations within a facility.

**Table 3 jcm-12-05105-t003:** Prescribing practice measured as percentage of respondents incorrectly selecting hypotonic fluids for each scenario question before and after the intervention, and the results of unadjusted and adjusted logistic regression odds ratio (OR) for selecting a hypotonic fluid post intervention compared to pre intervention. The OR was based on respondents from hospitals that participated both in the pre-intervention and post-intervention survey.

	Prescribing Practice of All Respondents				Prescribing Practice of Respondents from Hospitals That Participated Both in the Pre-Intervention and Post-Intervention Survey					
	Pre-Intervention (N = 201)		Post-Intervention (N = 154)		Pre-Intervention ^1^ (N = 164)		Post-Intervention ^1^ (N = 91)		Odds Ratio Unadjusted	Odds Ratio Adjusted
	*n*	(%)	*n*	(%)	*n*	(%)	*n*	(%)	(95% CI)	(95% CI)
Scenario 1 An otherwise healthy 18-year-old girl is hospitalised on suspicion of meningitis. She has thrown up and has diarrhoea. On examination, she appears pale with cold skin, a slightly increased heart rate, normal blood pressure and with decreased level of consciousness (Glasgow Coma Scale = score 14). Laboratory tests are normal.										
**Incorrect hypotonic fluid selected**	50	(24.9)	47	(30.5)	35	(21.3)	25	(27.5)	1.4 (0.8–2.5)	0.8 (0.4–1.6) ^2^
Scenario 2 A 5-year-old boy arrives at the emergency department with a head injury after falling from a bike. He is experiencing a headache and nausea, but no vomiting or signs of hypovolaemia. He has been unconscious for half an hour; however, the CT scan, clinical examination and laboratory results are all normal.										
**Incorrect hypotonic fluid selected**	59	(29.4)	39	(25.3)	54	(32.9)	21	(23.1)	0.6 (0.3–1.1)	0.7 (0.4–1.3) ^3^
Scenario 3 A 75-year-old woman arrives at the emergency department with a hip fracture after a fall. There are no signs of head injury. The patient has had a poor appetite for a long time. Medical history includes thiazide diuretics for hypertension, but otherwise she is healthy. Clinical examination shows symptoms of hypovolaemia: cold and pale skin, heart rate at 100 bpm and a slightly increased respiratory rate. Laboratory tests show P-Na = 110 mmol/L.										
**Incorrect hypotonic fluid selected**	21	(10.4)	9	(5.8)	16	(9.8)	7	(7.7)	0.8 (0.3–1.9)	- ^4^
Scenario 4 A 28-year-old woman is hospitalised on suspicion of medication poisoning and large intake of water. She vomits and complains of headaches. She exhibits strange behaviour, has muscle rigidity and a Glasgow Coma Scale score of 14. ABC is normal. Arterial blood gas shows P-Na = 118 mmol/L.										
**Incorrect hypotonic fluid selected**	33	(16.4)	11	(7.1)	27	(16.5)	9	(9.9)	0.6 (0.3–1.2)	- ^4^
Sum of incorrect hypotonic fluid selected for all scenarios	163	(20.3)	106	(17.2)	132	(20.1)	62	(17.0)		

^1^ Only hospitals that participated in both the pre-intervention and post-intervention survey are included. ^2^ Adjusted for the following variables: gender, number of patients treated with intravenous fluids weekly, complexity (i.e., complexity of services the ED provide), and type of ED (i.e., type of patients they serve at the ED, e.g., children or adults). Complexity and size are correlated, and thus only complexity is included in the model. ^3^ Adjusted for the following variables: age and type of ED (i.e., type of patients they serve at the ED, e.g., children or adults). Age and position are correlated, and thus only age is included in the model. ^4^ No need for adjustment.

**Table 4 jcm-12-05105-t004:** Correct responses to knowledge questions pre-intervention and post-intervention^1^.

		Pre		Post	
		*n*	(%)	*n*	(%)
Q10: Which of the following sentences are correct? (*n_pre_^2^* = 198; *n_pos_^3^* = 152)		158	(79.8)	118	(77.6)
	Most often acutely ill patients in need of IV fluids have increased renal water excretion				
	**Most often acutely ill patients in need of IV fluids have decreased renal water excretion**				
	Most often acutely ill patients in need of IV fluids have normal renal water excretion				
Q11a: How will Darrow–glucose ([Na^+^] = 31 mmol/L) affect the P-Na in a patient with decreased water excretion? (*n_pre_* = 184; *n_pos_* = 141)		85	(46.2)	65	(46.1)
	Large increase in P-Na with a risk of sodium overload				
	Slight increase in P-Na				
	Unchanged				
	Slight decrease in P-Na				
	**Large decrease in P-Na with a risk of hyponatraemia**				
Q11b: How will Potassium–sodium–glucose ([Na^+^] = 40 mmol/L) affect the P-Na in a patient with decreased water excretion? (*n_pre_* = 179; *n_post_* = 140)		56	(31.3)	43	(30.7)
	Large increase in P-Na with a risk of sodium overload				
	Slight increase in P-Na				
	Unchanged				
	Slight decrease in P-Na				
	**Large decrease in P-Na with a risk of hyponatraemia**				
Q11c: How will Ringer’s lactate ([Na^+^] = 130 mmol/L) affect the P-Na in a patient with decreased water excretion? (*n_pre_* = 179; *n_post_* = 137)		43	(24.0)	51	(37.2)
	Large increase in P-Na with a risk of sodium overload				
	Slight increase in P-Na				
	Unchanged				
	**Slight decrease in P-Na**				
	Large decrease in P-Na with a risk of hyponatraemia				
Q11d: How will 0.9% NaCl with 5% glucose ([Na^+^] = 154 mmol/L) affect the P-Na in a patient with decreased water excretion? (*n_pre_* = 177; *n_post_* = 134)		55	(31.1)	43	(32.1)
	Large increase in P-Na with a risk of sodium overload				
	Slight increase in P-Na				
	**Unchanged**				
	Slight decrease in P-Na				
	Large decrease in P-Na with a risk of hyponatraemia				
Q12: In case of increased blood sugar (above 12 mmol/L), the measured plasma sodium (P-Na) must be corrected because the measured P-Na is (*n_pre_* = 173; *n_post_* = 133):		52	(30.1)	56	(42.1)
	**Falsely low**				
	Falsely high				
	There is no need for correction				
Q13: Which of the following diseases/symptoms may be indicative of potentially increased intracranial pressure (ICP)? (*n_pre_* = 164; *n_post_* = 130)		23	(14.0)	14	(10.8)
	**Meningitis**				
	Shortness of breath				
	**Concussion**				
	Chest pain				
	**Seizure**				
	**Acute liver failure**				
	Acute abdomen				
	Hip fracture				
Q14: Which of the following symptoms are indicative of severe symptoms of hyponatraemia and require acute treatment of hyponatraemia?^4^ (*n_pre_* = 163; *n_post_* = 130)		103	(63.2)	81	(62.3)
	**Altered level of consciousness**				
	**Seizure**				
	Infection				
	Chest pain				
	Anaemia				
Sum of correct responses to all knowledge questions		541	(38.2)	471	(42.9)

^1^ Percentages of correct responses for knowledge questions are based on those who answered the specific question. The total number of respondents is indicated for each question in brackets. Correct responses are marked in bold. All questions included an option of ‘do not know’, which is not shown in the table. ^2^ *n_pre_* = number of respondents in the pre-intervention survey. ^3^ *n_post_* = number of respondents in the post-intervention survey. ^4^ ‘Muscle rigidity’ was included in the pre-intervention questionnaire but has been removed as it is not mentioned as a severe symptom of hyponatraemia in the current guideline on hyponatraemia [10].

## Data Availability

Data are available on request.

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
