# Peer review of "Impact of a Treatment Guide on Intravenous Fluids in Minimising the Risk of Hospital-Acquired Hyponatraemia in Denmark"

_jcm, 2023, doi:10.3390/jcm12155105_

Round 1
Reviewer 1 Report
The authors address the important topic of provider education on hypotonic fluid administration via a pre/post-intervention survey study. While the survey questions were developed carefully and an analysis was performed, the study is limited by a very low intervention rate. Please see further comments below. Thank you for the opportunity to review this work.
Page 2, Line 76: “The post-intervention survey started one year after the first distribution of the treatment guide” – shouldn’t this be 11 months? Treatment guide in June, survey in May.
Page 4, Line 137: Please be clearer at the beginning of this section regarding the primary outcome. You clarify this more at the end of the methods section but I would push this explanation to this primary outcome section.
Page 4, Line 160: What were the actual questions that remained in the post-intervention survey?
Page 4, Line 179: Please briefly state in this methods section what types of variables were used for univariable analysis. You state them later in the Results section but these should be introduced in the Methods section.
Page 5, Line 204: “Older physicians (>34 years), more senior physicians” is this a typo? I don’t understand the syntax here, are the older physicians and more senior physicians the same group?
Table 2: Can you please clarify what a junior doctor is? It appears that a PhD was included in the survey, do they have prescribing privileges for IV fluids?
Table 3: Please clarify which variables were inputted into the model for this table? What exactly do complexity and type of hospital mean? An initial table showing the factors associated (and not associated) with incorrect answers is necessary. For example, what was the OR and p-value for years of experience and incorrect answers?
Table 4: The methods section states that 5 of the 8 questions were removed to improve the response rate in the post-intervention survey, but it looks like all 8 questions were still included here.
Reviewer 2 Report
The study focused on hypotonic intravenous (IV) fluids are associated with an increased risk of hospital-acquired hyponatraemia eventually leading to brain injury and death. The evaluated the effectiveness of a treatment guide to improve prescribing practices of IV fluids. The study gives a nice overview of the hyponatraemia and related issues associated with it.
Author Response
Dear reviewer,
Thank you for taking the time to review our study and for the positive feedback.
Sincerely, on behalf of the authors,
Per Sindahl
Reviewer 3 Report
The manuscript by Sindahl et al. describes the impact of a treatment guide on intravenous fluids to minimize the risk of hospital-acquired hyponatraemia in Denmark. The article is original and well written. However, I have some issues to be addressed.
- The introduction should begin with a brief paragraph explaining the importance of fluid therapy in hospital (doi: 10.1093/bja/aeu300), mentioning the strategies in use to avoid fluid overload, such as the use of passive leg raising (doi: 10.1007/s00134-015-4134-1), inferior vena cava assessment (doi: 10.1016/j.jcrc.2022.154108) and fluid challenge (doi: 10.1186/s13054-022-04056-3). Please discuss and add these 4 references.
- - The fact that the treatment guide and the surveys were only administered to physicians working in Danish Emergency departments should also be underlined as a limitation of the study.
Round 2
Reviewer 1 Report
The authors have addressed my concerns regarding their work.